

# Intervention effects of five cations and their correction on hemolytic activity of tentacle extract from the jellyfish *Cyanea capillata*

Hui Zhang*, Qianqian Wang*, Liang Xiao and Liming Zhang

Department of Marine Biotechnology, Faculty of Naval Medicine, Second Military Medical University, Shanghai, China
* These authors contributed equally to this work.

## ABSTRACT

Cations have generally been reported to prevent jellyfish venom-induced hemolysis through multiple mechanisms by spectrophotometry. Little attention has been paid to the potential interaction between cations and hemoglobin, potentially influencing the antagonistic effect of cations. Here, we explored the effects of five reported cations, $La^{3+}$, $Mn^{2+}$, $Zn^{2+}$, $Cu^{2+}$ and $Fe^{2+}$, on a hemolytic test system and the absorbance of hemoglobin, which was further used to measure their effects on the hemolysis of tentacle extract (TE) from the jellyfish *Cyanea capillata*. All the cations displayed significant dose-dependent inhibitory effects on TE-induced hemolysis with various dissociation equilibrium constant ($K_d$) values as follows: $La^{3+}$ 1.5 mM, $Mn^{2+}$ 93.2 mM, $Zn^{2+}$ 38.6 mM, $Cu^{2+}$ 71.9 μM and $Fe^{2+}$ 32.8 mM. The transparent non-selective pore blocker $La^{3+}$ did not affect the absorbance of hemoglobin, while $Mn^{2+}$ reduced it slightly. Other cations, including $Zn^{2+}$, $Cu^{2+}$ and $Fe^{2+}$, greatly decreased the absorbance with $K_d$ values of 35.9, 77.5 and 17.6 mM, respectively. After correction, the inhibitory $K_d$ values were 1.4 mM, 45.8 mM, 128.5 μM and 53.1 mM for $La^{3+}$, $Zn^{2+}$, $Cu^{2+}$ and $Fe^{2+}$, respectively. $Mn^{2+}$ did not inhibit TE-induced hemolysis. Moreover, the inhibitory extent at the maximal given dose of all cations except $La^{3+}$ was also diminished. These corrected results from spectrophotometry were further confirmed by direct erythrocyte counting under microscopy. Our results indicate that the cations, except for $La^{3+}$, can interfere with the absorbance of hemoglobin, which should be corrected when their inhibitory effects on hemolysis by jellyfish venoms are examined. The variation in the inhibitory effects of cations suggests that the hemolysis by jellyfish venom is mainly attributed to the formation of non-selective cation pore complexes over other potential mechanisms, such as phospholipases A2 (PLA2), polypeptides, protease and oxidation. Blocking the pore-forming complexes may be a primary strategy to improve the in vivo damage and mortality from jellyfish stings due to hemolytic toxicity.

Corresponding authors
Liang Xiao,
hormat830713@hotmail.com
Liming Zhang,
lmzhang@smmu.edu.cn

## INTRODUCTION

Jellyfish are free-swimming marine animals consisting of a gelatinous umbrella-shaped bell and trailing tentacles. While they are found in coastal water zones worldwide, jellyfish populations fluctuate greatly in accordance with ocean climate and, perhaps, other factors related to human interactions (*Williamson et al., 1984*; *Winter et al., 2010*). Jellyfish range from about 1 mm to nearly 2 m in bell height and diameter; the tentacles that rim the umbrella typically extend beyond their bell dimension. Contact with jellyfish tentacles, even when beached and dying, can trigger millions of nematocysts to pierce the skin and inject venom through inverted long spiny tubules, thereby causing toxic manifestations from no effect to extreme pain and even death (*Carrette & Seymour, 2004*).

The jellyfish venom in the nematocyst, similarly to many other types of venom, is a complex mixture of bioactive proteins and peptides that have demonstrated a wide spectrum of biological activities (*Bloom, Burnett & Alderslade, 1998*; *Chung et al., 2001*; *Yanagihara et al., 2002*; *Sanchez-Rodriguez, Torrens & Segura-Puertas, 2006*; *Brinkman & Burnell, 2008*), including dermonecrotic, cardiotoxic, neurotoxic, hemolytic, enzymatic, immunogenic and inflammatory effects. Despite over 50 years of research, the pathophysiological processes and mechanisms of the toxic proteins and peptides in this venom have yet to be elucidated. In general, the main cause of death is believed to be cardiotoxicity, while the hemolytic activity is considered a preliminary damage factor and offers an approach to disentangle the complex venom. However, it is reported that hemolysis can range from simple nuisance to serious pathological and lethal events, and is a frequent effect of a number of jellyfish venoms acting as lytic protein/peptides that alter cell permeability resulting in ion transport, cell swelling and osmotic lysis, whereas others are phospholipases inducing degradation of bilayer phospholipids or channel-forming agents embedded into the membrane (*Mariottini, 2014*).

Determining the optical absorbance of released hemoglobin from lysed erythrocytes at 414 or 545 nm by spectrophotometry is a simple method to test the hemolytic activity of jellyfish venom. This method is widely utilized in cytolytic activity evaluation, hemolytic compound purification and identification, and the identification of inhibitors necessary to explore the hemolytic mechanism and develop novel anti-hemolytic strategies. Dozens of compounds, including osmotic protectants, lipids, proteases, antioxidants and cations, have demonstrated the potential to prevent hemolysis induced by jellyfish venom by either impeding the destruction of membrane lipids or hindering the formation of pores on the cell membrane (*Li et al., 2005*; *Marino et al., 2008*; *Marino, Morabito & La Spada, 2009*; *Morabito et al., 2014*). Compared with various types of inhibitors, certain cations, such as $La^{3+}$ (*Bailey et al., 2005*), $Mn^{2+}$ (*Li et al., 2005*), $Zn^{2+}$ (*Yu et al., 2007*), $Cu^{2+}$ (*Marino, Morabito & La Spada, 2009*) and $Fe^{2+}$ (*Yu et al., 2007*), are usually effective at millimolar concentrations, a relatively high dose that partially precipitates the jellyfish proteins, such as collagen and other toxic compounds. Cations that are colored in aqueous solutions, such as $Cu^{2+}$ and $Fe^{2+}$, may interfere with the absorbance of hemoglobin by spectrophotometry, raising concerns regarding whether

this testing method reveals the true inhibitory effects of cations on the hemolytic activity of jellyfish venom. In the current study, we aim to determine the intervention effects of cations $La^{3+}$, $Mn^{2+}$, $Zn^{2+}$, $Cu^{2+}$ and $Fe^{2+}$ on the hemolytic activity of jellyfish venom and to further exploit its possible mechanism underlying the hemolytic activity of jellyfish venom.

## MATERIALS AND METHODS

### Preparation of tentacle extract from the jellyfish *Cyanea capillata*

Specimens of *Cyanea capillata* collected in June 2015 in the Sanmen Bay, East China Sea, were identified by Professor Huixin Hong from the Fisheries College of Jimei University, Xiamen, China. The isolated tentacles were placed in plastic bags with dry ice and immediately shipped to Shanghai, where the samples were stored in a $-80\ °C$ freezer until use. The tentacle extract (TE) preparation procedure has been described in previous reports (*Wang et al., 2013a*, *2013b*, *2013c*, *2013d*, *2014*; *Zhang et al., 2014*). Briefly, tentacles at $-80\ °C$ were thawed and immersed in seawater (28 g/L NaCl, 5 g/L $MgCl_2 \cdot 6H_2O$, 0.8 g/L KCl, 1.033 g/L $CaCl_2$) to allow tissue autolysis and stirred gently for four days. The autolyzed mixture was then centrifuged at 10,000$g$ for 15 min, and the resultant supernatant liquid was collected as the TE. The TE was dialyzed against PBS (pH 7.4, 0.01 M) for 8 h before use. All the procedures were performed at 4 °C.

### Erythrocyte suspension

Samples of blood were drawn through the tail vein of male Kunming mice (3–6 months) that were purchased from the Laboratory Animal Center of the Second Military Medical University (SMMU). The mice were provided with sufficient food and water, and all animal handlings were approved by the SMMU Ethics Committee. Fresh heparinized mouse blood (100 μL) was suspended in 10 mL 0.01 M phosphate buffer containing 0.9% NaCl (pH = 7.35, 300 $mOsm/kgH_{20}$). To prepare a pure erythrocyte suspension, the diluted blood sample was centrifuged at 1,000$g$ for 10 min. The supernatant buffy coat and blood serum were discarded, and the erythrocyte pellet was washed twice and suspended in the same buffer to a final concentration of 0.45% (v/v) (*Kang et al., 2009*; *Li et al., 2013*; *Wang et al., 2013d*).

### Hemoglobin solution

In total, 10 mL of erythrocyte suspension were transferred to a 15 mL centrifugal tube and subjected to a series of sonication periods. After sonication for 60 s in total, the samples were allowed to cool for 10 s between ultrasound pulses, using a Misonix S-4000 sonicator (Qsonica, Newtown, CT, USA) set to 20 kHz and 25 W. The sonicated erythrocyte sample was centrifuged at 10,000$g$ for 30 min to remove the fragmentized cell membrane and released organelles, and the resultant supernatant was hemoglobin solution.

### Hemolytic test by spectrophotometry

The hemolytic activity of TE was first tested by spectrophotometry. Various concentrations of TE (30, 90, 180, 270, 360, 450 and 540 μg/mL) were added to the

 

erythrocyte suspension (100 μL, 0.45% in 0.01 M phosphate buffer containing 0.9% NaCl, pH = 7.35, 300 mOsm/kgH$_{20}$). The total volume of the test system was 200 μL. The samples were incubated at 37 °C for 30 min in a water bath accompanied by mild horizontal shaking. The intact erythrocytes and erythrocyte ghosts were removed by centrifugation at 2,000$g$ for 5 min. A 150 μL portion of the supernatant fluid was transferred to a 96-well microplate, and its optical absorbance (OD) was measured at 415 nm by spectrophotometry. The concentration of the released hemoglobin from the lysed erythrocytes was taken as the index of the TE-induced hemolysis. The negative (0.01M phosphate buffer) and positive (30 μg/mL saponin) controls were taken as 0% and 100% hemolysis, respectively. The hemolytic activity of TE was expressed as % absorbance, compared to the positive control group.

## Cation interventions

Different amounts of LaCl$_3$ (1.2, 1.4, 1.6 and 1.8 mM), MnCl$_2$ (20, 40, 50, 100 and 400 mM), ZnCl$_2$ (10, 20, 40 and 100 mM), CuCl$_2$ (30, 60, 90 and 120 μM) and FeSO$_4$ (20, 40, 80 and 120 mM) dissolved in 0.01 M phosphate buffer were added to an erythrocyte suspension (100 μL, 0.45% in 0.01 M phosphate buffer containing 0.9% NaCl, pH = 7.35, 300 mOsm/kgH$_{20}$), followed by the addition of 360 μg/mL TE, to test their inhibitory effect on TE-induced hemolysis. Similarly, the same cation solutions were added to 100 μL of hemoglobin suspension to determine the effect of the cations on the absorbance of hemoglobin at 414 nm. A Nanodrop 1000 (Thermo, Waltham, MA, USA) was used to measure the UV–Vis spectra of hemoglobin solution ranging from 220 to 750 nm in the presence of Cu$^{2+}$ (30, 60 and 120 μM) and Fe$^{2+}$ (20, 40 and 120 mM).

## Confocal laser scanning microscopy

Direct erythrocyte counting was also carried out using confocal laser scanning microscopy (CLSM). To determine the effect of cations on TE-induced hemolysis, LaCl$_3$ (1.8 mM), MnCl$_2$ (400 mM), ZnCl$_2$ (100 mM), CuCl$_2$ (120 μM) or FeSO$_4$ (120 mM) were added to the erythrocyte solution, followed by 360 μg/mL TE. After incubation at 37 °C for 30 min in a water bath, accompanied by mild horizontal shaking, the erythrocyte suspension was mixed and transferred to a confocal dish for cell counting.

## Data analysis

All the quantitative data are expressed as the mean ± SD. Statistical analyses were performed using one-way ANOVA and followed by Student–Newman–Keuls test with the SPSS 22.0 software. The pictures were depicted with the Origin software. A 0.05 level of probability was used as the level of significance.

## Ethical statement

The investigation was carried out in conformity with the requirements of the Ethics Committee of the Second Military Medical University and National Institutes of Health (NIH) guide for care and use of Laboratory animals (NIH Publications No. 8023). Jellyfish catching was permitted by the East China Sea Branch, State Oceanic Administration, People's Republic of China.

## RESULTS

### Dose-dependent TE hemolysis by spectrophotometry

Using current spectrophotometric methods, we examined the dose-dependent relationship of hemolysis by TE from the jellyfish *Cyanea capillata*. Figure 1 shows a dose–response curve depicted by the Hill equation within the TE dose of 540 µg/mL. The dissociation equilibrium constant ($K_d$) value (*Tomasi et al., 2013*; *Dupin et al., 2015*) was 176.2 ± 8.2 µg/mL. The hemolytic curve slowly increased up to 90 µg/mL, reaching values of less than 10%, and then sharply increased to 360 µg/mL, reaching the value of approximately 90%. When the exposure concentrations were greater than 360 µg/mL, the curve began to reach the maximal plateau. Thus, for all future experiments, we uniformly utilized the TE concentration of 360 µg/mL to evaluate TE hemolysis.

### Inhibitory effects of cations on TE-induced hemolysis by spectrophotometry

Several studies (*Rosenthal et al., 1990*; *Edwards & Hessinger, 2000*; *Bailey et al., 2005*) have previously reported that the cations $La^{3+}$, $Mn^{2+}$, $Cu^{2+}$, $Zn^{2+}$ and $Fe^{2+}$ antagonize hemolysis induced by nematocyst venom or jellyfish TE by either blocking the venom-formed membrane pore or stabilizing the cell membrane. We, therefore, examined the anti-hemolytic effects of the above cations on TE-induced hemolysis using traditional spectrophotometric methods. As expected, all cations displayed significant dose-dependent anti-hemolytic effects on TE-induced hemolysis, although with differing inhibitory concentrations and $K_d$ values (Figs. 2A–2E). The effective inhibitory concentrations of the non-specific ion channel blocker $La^{3+}$ were between 1 and 1.8 mM, with a $K_d$ value 1.5 ± 0.01 mM. $Cu^{2+}$ displayed an even stronger suppression of TE-induced hemolysis; its inhibitory concentration was less than 120 µM, and the $K_d$ value was 71.9 ± 6.5 µM. Other bivalent cations, including $Mn^{2+}$, $Zn^{2+}$ and $Fe^{2+}$, displayed relatively weaker suppressive effects on TE-induced hemolysis compared with $La^{3+}$ and $Cu^{2+}$. Their ranges of inhibitory concentrations were ~400, ~120 and ~100 mM, with $K_d$ values of 93.2 ± 7.0 mM, 38.6 ± 36.3 mM and 32.8 ± 76.1 mM, respectively. Moreover, to show the anti-hemolysis effects of cations more distinctly, we compared the respective maximal suppressive extents of the five cations with TE treatment group (Fig. 2F). According to the spectrophotometric values, $La^{3+}$, $Cu^{2+}$, $Zn^{2+}$ and $Fe^{2+}$ resulted in complete inhibitions of less than 10%, while $Mn^{2+}$ showed partial suppression with a maximum inhibition of approximately 40%. Therefore, our results showed that the inhibitory effects of cations on TE-induced hemolysis were in accordance with previous investigations.

### Influences of cations on the hemolytic test system by spectrophotometry

In the process of spectrophotometric hemolysis determination, we observed that some cations resulted in colored aqueous solutions, such as $Cu^{2+}$ (green), $Fe^{2+}$ (aqua) and $Mn^{2+}$ (red); in addition, the $Zn^{2+}$ solution was slightly ivory as a result of the high

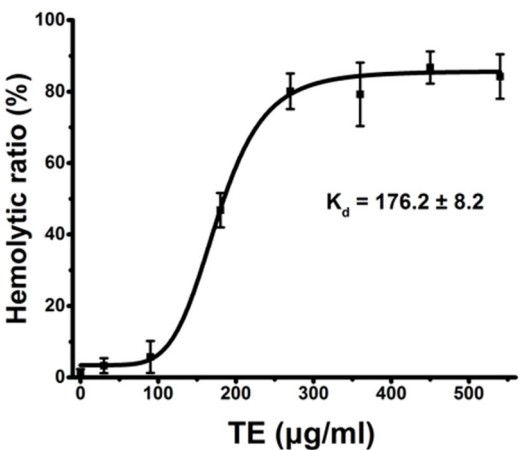

**Figure 1 Hemolysis ratio (%) after 30 min of treatment with varying TE concentrations.** The dose–response curve is depicted based on Hill's co-operation analysis. All data are presented as the mean ± SD ($n = 3$).

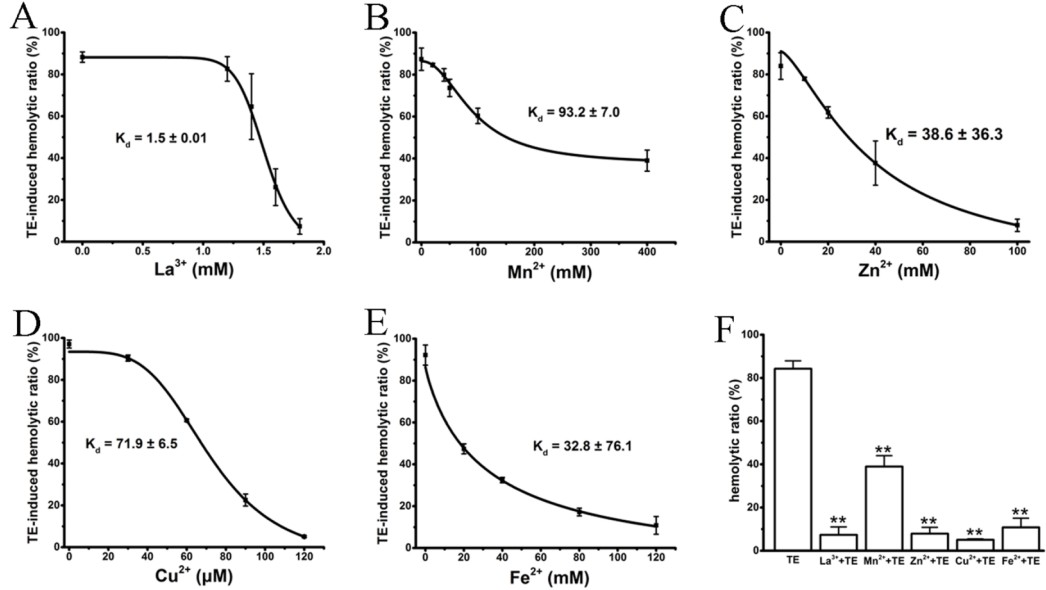

**Figure 2 Intervention of various cations on TE-induced hemolysis.** The cations $La^{3+}$ (A); $Mn^{2+}$ (B); $Zn^{2+}$ (C); $Cu^{2+}$ (D) and $Fe^{2+}$ (E) were pre-incubated with a 0.45% erythrocyte suspension for 30 min before the administration of 360 $\mu g/mL$ TE. The corresponding dose–response curves are depicted based on Hill's co-operation analysis. (F) Comparison of the changes of TE-induced hemolysis ratio (%) under the maximum anti-hemolytic concentrations $La^{3+}$ (1.8 mM); $Mn^{2+}$ (400 mM); $Zn^{2+}$ (100 mM); $Cu^{2+}$ (120 $\mu M$) and $Fe^{2+}$ (120 mM). All data are demonstrated as the mean ± SD ($n = 3$). $^{*}p < 0.05$, compared to the TE-treatment group.

concentration-induced hydrolysis. Although erythrocytes and hemoglobin exhibit specific absorbance peaks at 414 and 545 nm, we were interested whether these colorful cations influenced the test system, thereby complicating an analysis of their effects on TE-induced hemolysis. We first compared the absorbance values of the colored cation solutions at their maximum anti-hemolytic concentrations at 415 nm (Fig. 3A).

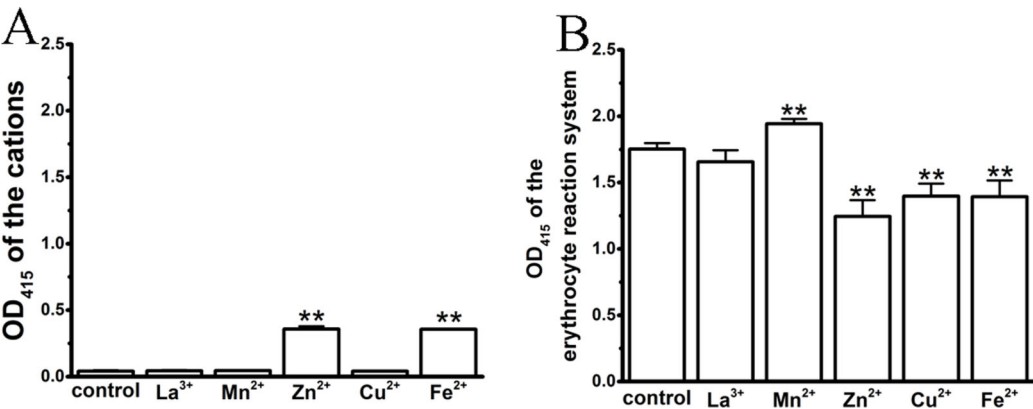

**Figure 3 Effects of the cations on an erythrocyte reaction system by spectrophotometry, where the maximal anti-hemolytic concentrations of the cations were used.** (A) Optical absorbance (OD) of cations at 415 nm. Control (0.9% NaCl), $La^{3+}$ (1.8 mM), $Mn^{2+}$ (400 mM), $Zn^{2+}$ (100 mM), $Cu^{2+}$ (120 μM) and $Fe^{2+}$ (120 mM). (B) Optical absorbance (OD) of the erythrocyte reaction system at 415 nm upon treatment with cations. Control (0.9% NaCl); $La^{3+}$ (1.8 mM); $Mn^{2+}$ (400 mM); $Zn^{2+}$ (100 mM); $Cu^{2+}$ (120 μM) and $Fe^{2+}$ (120 mM). All data are demonstrated as the mean ± SD ($n = 3$). $^*p < 0.05$, compared to the control group.

$La^{3+}$, $Mn^{2+}$ and $Cu^{2+}$ displayed similar values to that of control, while $Zn^{2+}$ and $Fe^{2+}$ showed small increases. We then tested the effects of the cations on the 0.45% erythrocyte solutions (Fig. 3B). As expected, the colorful $Zn^{2+}$, $Cu^{2+}$ and $Fe^{2+}$ cations resulted in a decrease in the absorbance values of the erythrocytes, while $Mn^{2+}$ increased the absorbance values and the transparent $La^{3+}$ had little effect, suggesting that the cation colors might interfere with the absorbance spectrum of hemoglobin.

## Effect of cations on the hemoglobin solution

We tested the effects of cations, at the same concentrations used above, on released hemoglobin from a 0.45% erythrocyte suspension (Fig. 4). We did not see any effect by the transparent $La^{3+}$ on the hemoglobin solutions (Fig. 4A). The $Mn^{2+}$ solution displayed a small decrease in the absorbance values (Fig. 4B). The colorful $Zn^{2+}$, $Cu^{2+}$ and $Fe^{2+}$ greatly diminished the absorbance values of hemoglobin at the given concentrations; the resulting $K_d$ values were 35.9 ± 25.6 mM, 77.5 ± 7.7 μM and 17.6 ± 66.4 mM, respectively (Figs. 4C–4E). $Cu^{2+}$ is the only one of the five cations that was used in μM/L and $Fe^{2+}$ has a millimolar concentration in blood, so we measured the UV–Vis spectra of hemoglobin solution in the presence of $Cu^{2+}$ (30, 60 and 120 μM) and $Fe^{2+}$ (20, 40 and 120 mM) (Fig. 5). The results showed that, at 415 nm where hemoglobin has a peak absorption, $Cu^{2+}$ (30, 60 and 120 μM) (Fig. 5A) and $Fe^{2+}$ (120 mM) (Fig. 5B) obviously decreased the absorbance value of hemoglobin. These experiments indicate that the colorful cations can change the absorbance values of erythrocytes and hemoglobin at 415 nm, thereby influencing the hemolytic test system.

## Correction of the inhibitory effects of cations on TE-induced hemolysis

Therefore, the hemolytic effects of the colored cations can be separated into two effects. The first is the true inhibition of TE-induced hemolysis, and the other is a false positive

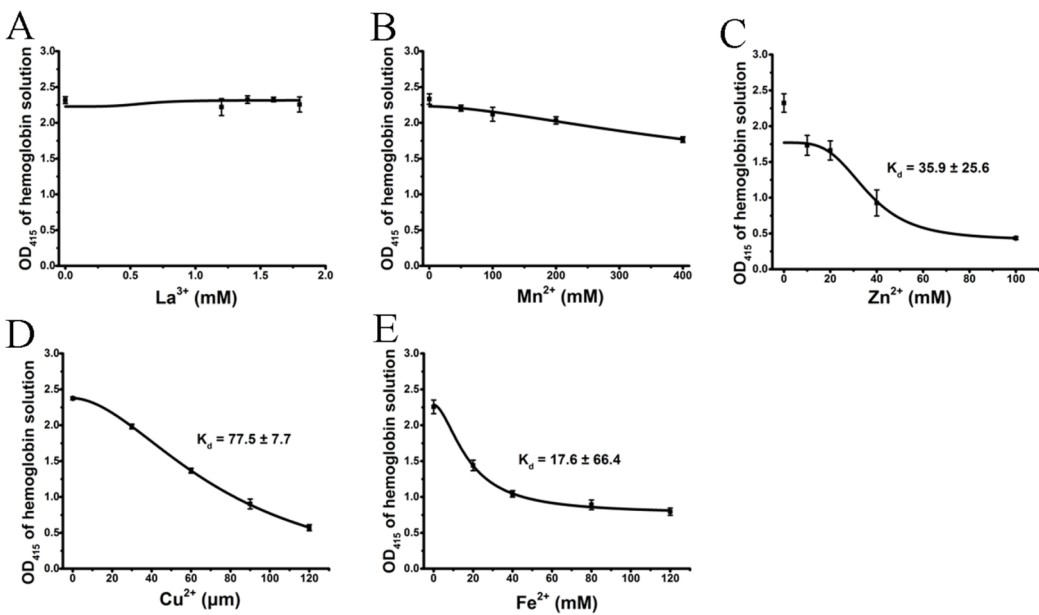

**Figure 4 Effects of the cations on a hemoglobin solution by spectrophotometry.** The hemoglobin solution was obtained by sonicating a 0.45% erythrocyte suspension. $La^{3+}$ (A); $Mn^{2+}$ (B); $Zn^{2+}$ (C); $Cu^{2+}$ (D) and $Fe^{2+}$ (E) are displayed. Corresponding dose–response curves are depicted based on Hill's co-operation analysis. All data are demonstrated as the mean ± SD ($n = 3$).

effect caused by the modulation of the hemoglobin absorbance by the colored cation solution. Because the absorbance values of hemoglobin were modified in proportion to the cation concentration, and every tested concentration was determined, we used the equation "$y = ax$" to adjust the hemolytic ratio at every given concentration, where "$y$" is the real hemolytic ratio, "$x$" is the determined hemolytic ratio and "$a$" is an adjustment coefficient that is the inverse of the ratio of cations at the corresponding concentration. Except for the transparent $La^{3+}$ (Fig. 6A), the anti-hemolysis curves of four other cations were right-shifted. The corrected $K_d$ values were 1.4 ± 6.5 mM, 45.8 ± 15 mM, 128.5 ± 130 μM and 53.1 ± 111.3 mM for $La^{3+}$, $Zn^{2+}$, $Cu^{2+}$ and $Fe^{2+}$, respectively (Fig. 6). To our surprise, $Mn^{2+}$ did not show any inhibition after correction (Fig. 6B). Meanwhile, all cations at the given maximal concentrations displayed much smaller inhibitory effects on TE-induced hemolysis although the differences were still significant. Thus, the correction indicated that the anti-hemolytic effects of cations $Mn^{2+}$, $Cu^{2+}$, $Zn^{2+}$ and $Fe^{2+}$ (Figs. 6B–6E) on TE-induced hemolysis were over-estimated and must be corrected using their respective cation–hemoglobin concentration curves.

## Effects of the cations on TE-induced hemolysis under microscopy

To further confirm our results, we counted the erythrocytes under a microscope after the co-incubation with cations alone or cations in the presence of TE, and compared the erythrocyte numbers before and after TE treatment to give the "Erythrocyte amount (%)" (Fig. 7). The number of erythrocytes was significantly decreased in the TE group, and their shape was changed from the typical biconcave discoid to convex and rounded. In all cation groups, the internal cations did not change the number or the shapes of the

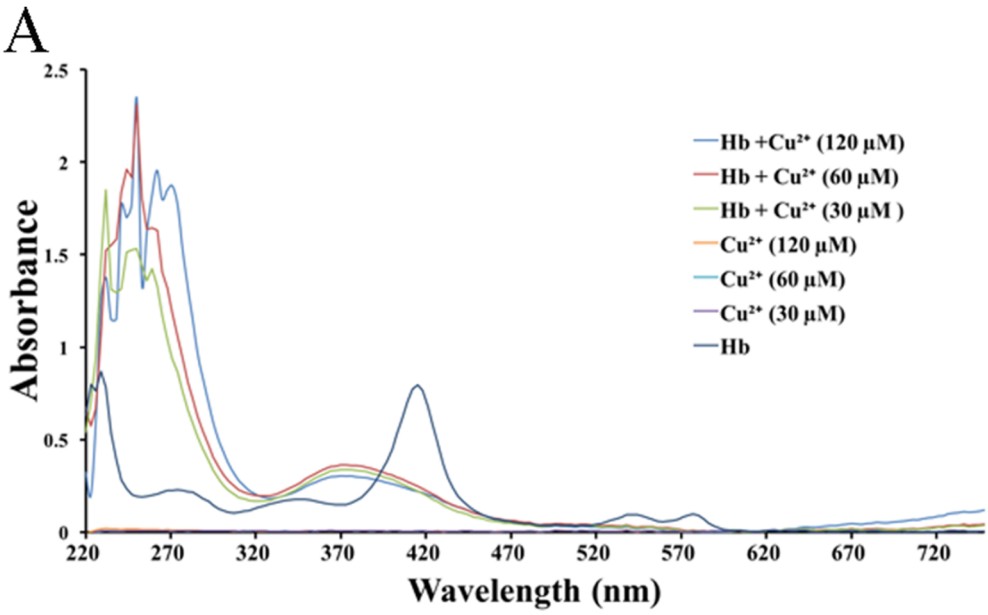

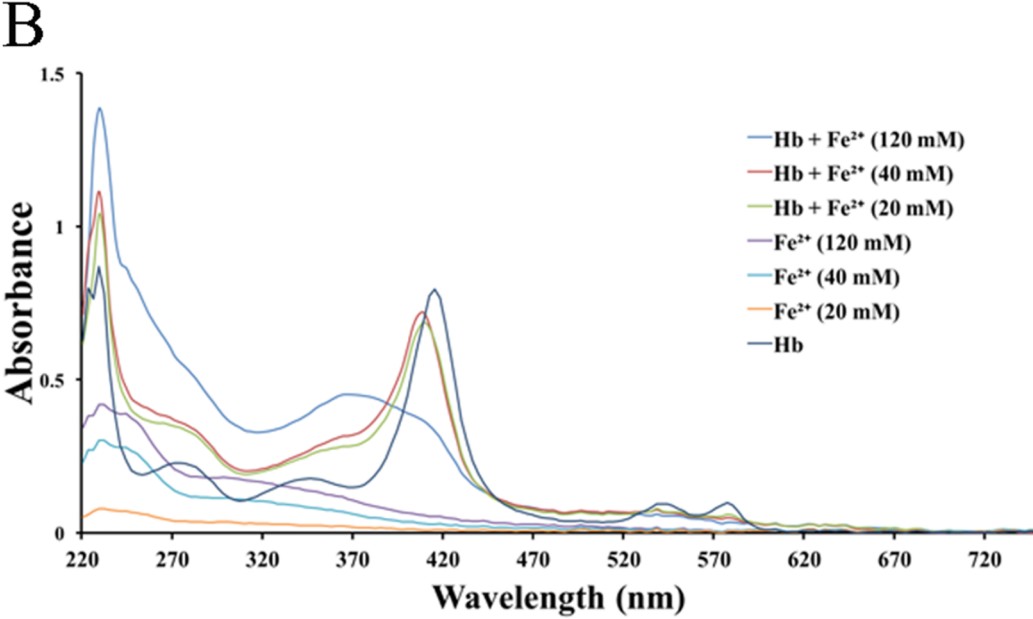

**Figure 5 UV–Vis spectroscopy of hemoglobin solution.** (A) UV–Vis spectroscopy of hemoglobin solution in the presence of $Cu^{2+}$ (30, 60 and 120 μM). (B) UV–Vis spectroscopy of hemoglobin solution in the presence of $Fe^{2+}$ (20, 40 and 120 mM).

erythrocytes. To our surprise, a large amount of floccule was observed encircling the erythrocytes, especially in the $La^{3+}$, $Zn^{2+}$, $Fe^{2+}$ and $Cu^{2+}$ groups, which might be due to the interaction of cations with the collagen in jellyfish TE. In the $La^{3+}$, $Zn^{2+}$, $Fe^{2+}$ and $Cu^{2+}$ groups, more erythrocytes were present than in the TE-alone positive control group. No change in the erythrocyte amount was observed in the $La^{3+}$ group, suggesting the obvious inhibitory effect of $La^{3+}$ on TE-induced hemolysis. $Cu^{2+}$, $Zn^{2+}$ and $Fe^{2+}$ also displayed strong anti-hemolytic effects, while the erythrocyte count in the $Mn^{2+}$ group

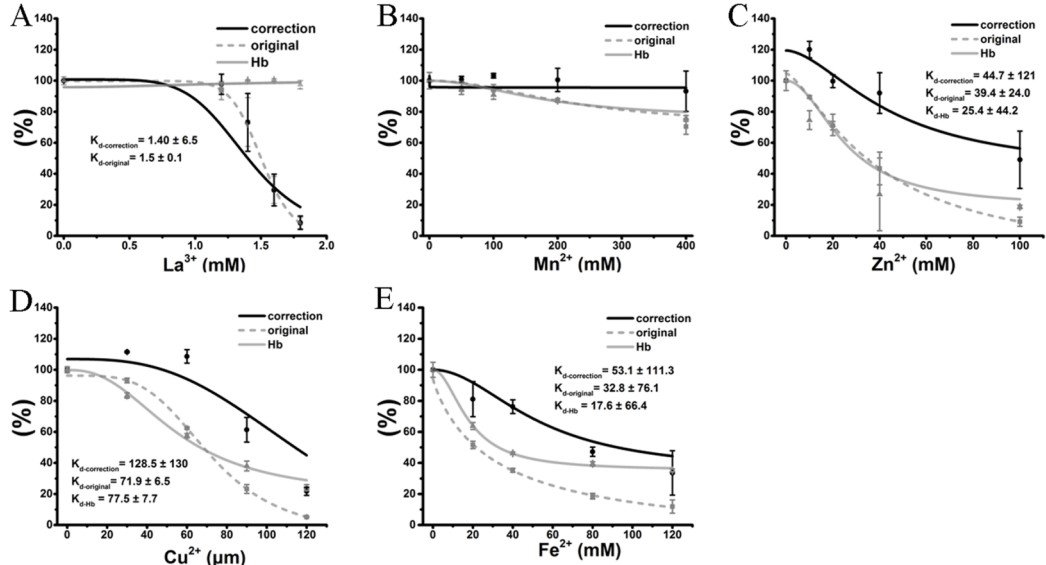

**Figure 6 Correction of cation inhibitory effects on TE-induced hemolysis.** $La^{3+}$ (A); $Mn^{2+}$ (B); $Zn^{2+}$ (C); $Cu^{2+}$ (D) and $Fe^{2+}$ (E) are displayed. The curves marked "correction" were adjusted according to the equation "$y = ax$", where "$y$" is the real hemolytic ratio, "$x$" is the determined hemolytic ratio and "$a$" is the adjustment coefficient, i.e., the inverse of the ratio of cations at the corresponding concentration on the absorbance values of hemoglobin. The curves marked "original" were adjusted according to the determined hemolytic ratio (Fig. 2). The curves marked "Hb" were adjusted according to the $OD_{415}$ of cations on the released hemoglobin from a 0.45% erythrocyte suspension by spectrophotometry (Fig. 4). $K_{d-correction}$, $K_{d-original}$, $K_{d-Hb}$ are all listed, respectively. Corresponding dose–response curves were depicted based on Hill's co-operation analysis. All data are demonstrated as the mean ± SD ($n = 3$).

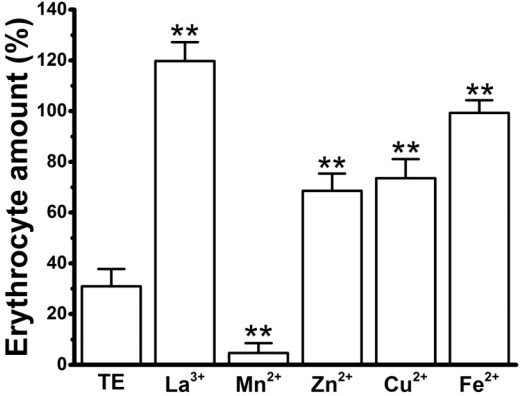

**Figure 7 Effects of the cations on TE-induced hemolysis by direct erythrocyte counting under microscopy using the maximal anti-hemolytic concentrations of cations [$La^{3+}$ (1.8 mM); $Mn^{2+}$ (400 mM); $Zn^{2+}$ (100 mM); $Cu^{2+}$ (120 μM) and $Fe^{2+}$ (120 mM)].** The erythrocyte amount (%) means the ratio of cell numbers in different cations plus TE groups versus cation-only groups. All data are demonstrated as the mean ± SD ($n = 3$). *$p < 0.05$, compared to the TE-treatment group.

was the lowest, indicating the weakest, or absence of, anti-hemolytic effects. These results indicate that the inhibitory effects of the cations on TE-induced hemolysis, as examined under microscopy, were roughly consistent with the determination by spectrophotometry after correction.

## DISCUSSION

We have examined the anti-hemolytic effects of five cations with reported inhibitory effects on the TE-induced hemolysis from the jellyfish *Cyanea capillata*. All the cations, except the transparent $La^{3+}$, were found to significantly influence the erythrocyte test system and the hemoglobin system by virtue of their colored aqueous solutions. The true antagonistic effects of the cations on TE-induced hemolysis were actually weaker than those tested by in previous spectrophotometry studies. For this study, corrected inhibitions were calculated from the effect of colored solutions on hemoglobin absorbance under the same conditions; these corrected values were supported by direct counting of erythrocytes under microscopy.

### Hemolytic determination and intervention

Hemolysis is a frequent effect of jellyfish stings. This dangerous condition is known to be caused by the jellyfish venom and can be lethal. Mature mammalian erythrocytes are highly differentiated cells that possess a large amount of hemoglobin without intracellular organelles, such as mitochondria and a nucleus; thus, the concentration of released hemoglobin at 414 nm (*Chung et al., 2001*; *Garcia-Arredondo et al., 2014*; *Kang et al., 2014*; *Morabito et al., 2014*) or 545 nm (*Kang et al., 2009*) is normally proportional to the number of lysed erythrocytes and reflects the hemolytic activity. This simple method for determination of hemolysis is widely used to compare hemolytic potencies, explore molecular mechanisms and search for potential interventions (*Long & Burnett, 1989*). Concentrations inducing 50% hemolysis by jellyfish venoms vary from "~ng/mL" to "~mg/mL" depending on the jellyfish species and sample extraction methods (*Mariottini, 2014*). The hemolytic activity of crude venom from isolated nematocysts of the Hawaiian box jellyfish *Carybdea alata* was 1–10 μg/mL and was reduced after exposure to proteolytic enzymes (trypsin, collagenase and papain), carbohydrates (D-lactulose and D-galactose, among others), EDTA and cations ($K^+$ and $Mn^{2+}$) but was increased by other cations, including $Mg^{2+}$, $Ca^{2+}$ and $Zn^{2+}$ (*Chung et al., 2001*). In another study, the hemolytic activity of the full venom of *Rhopilema esculentum* Kishinouye was 3.4 μg/mL and was affected by pH, temperature, EDTA, $(NH_4)_2SO_4$ and all tested divalent cations ($Mg^{2+}$, $Cu^{2+}$, $Zn^{2+}$, $Fe^{2+}$, $Ca^{2+}$ and $Mn^{2+}$) (*Yu et al., 2007*). Hemolysis by the crude venom of *Aiptasia mutabilis* was prevented by $Ca^{2+}$, $Ba^{2+}$ and $Cu^{2+}$, and suppressed to a minor extent by $Mg^{2+}$ and $K^+$ (*Marino, Morabito & La Spada, 2009*). In our previous study (*Lu et al., 2012*), the hemolytic activity of TE from the jellyfish *Cyanea capillata* was inhibited by $Mn^{2+}$, $Zn^{2+}$, $La^{3+}$, $Cu^{2+}$ and $Fe^{2+}$, while the hemolytic activity was increased in the presence of $K^+$, $Ca^{2+}$, $Mg^{2+}$ and $NH_4^+$.

Except for $K^+$, $Ca^{2+}$ and $Mg^{2+}$, it appears that other tested cations, including $Mn^{2+}$, $Zn^{2+}$, $La^{3+}$, $Cu^{2+}$ and $Fe^{2+}$, have an antagonistic effect on hemolysis by jellyfish venoms (*Lu et al., 2012*). Because traditional spectrophotometric methods test the absorbance of hemoglobin, which indirectly reflects the extent of erythrocyte hemolysis, cations interacting with hemoglobin or interfering with the absorbance of hemoglobin will lead to a false positive. Though the antagonist effects have been demonstrated to be reproducible,

the colors of some cations, such as $Cu^{2+}$ and $Fe^{2+}$, should lead researchers to be more cautious in judging their effects. Our initial results suggested that all tested cations, except for the transparent $La^{3+}$, significantly shifted the absorbance curves of hemoglobin to the right. Accordingly, the corrected antagonistic effects were right-shifted, except in the cases of $La^{3+}$ and $Mn^{2+}$. $La^{3+}$ displayed no effect on the curve, and $Mn^{2+}$ showed no antagonistic effect after correction, which came as a surprise to us. Notwithstanding the $K_d$ values, the maximum inhibitory values were also decreased in $Zn^{2+}$, $Cu^{2+}$ and $Fe^{2+}$. These results were further supported by the direct counting of erythrocytes using confocal microscopy.

## Hemolytic mechanism and compounds

Ever since hemolytic proteins were first purified and identified as CrTX-A and CrTX-B from the venom of the jellyfish *Carybdea rastoni* (*Nagai et al., 2000*), jellyfish hemolytic proteins have been developed as a novel family of taxonomically restricted cnidarian toxins (42–46 kDa) in the jellyfish species *Chironex fleckeri*, *Cyanea nozakii* Kishinouye, *Chironex yamaguchii* (as *Chiropsalmus quadrigatus*), and *Alatina moseri* (as *Carybdea alata*) (*Chung et al., 2001*; *Nagai et al., 2002*; *Brinkman et al., 2014*; *Mariottini, 2014*; *Strobos, 2014*). These hemolytic proteins were identified as pore-forming toxins by bioinformatics, leading to the hemolytic activity of jellyfish venom. In addition to the non-selective formation of pore complexes, at least the following four other factors have been shown to cause the hemolytic effects of jellyfish venom: (1) Protease and collagenase that are able to break the cell membrane via digestion of membrane proteins (*Li et al., 2014*, *2016*). (2) Phospholipase $A_2$ ($PLA_2$), the activity of which, in jellyfish venom, was discovered long ago. Several phospholipases have been identified recently using transcriptomic and proteomic analyses (*Nevalainen et al., 2004*; *Weston et al., 2013*; *Heo et al., 2016*). (3) Polypeptides: two novel cytolysins, designated oshem1 and oshem2, with respective molecular weights of 3 and 3.4 kDa, were identified from the tentacle of the Hydrozoan *Olindias sambaquiensis* (*Junior et al., 2014*). (4) Oxidizing compounds: we have previously reported that lipid peroxidation is a potential mechanism besides pore formation underlying hemolysis by TE from the jellyfish *Cyanea capillata* (*Ayed et al., 2011*; *Wang et al., 2013d*). Therefore, the hemolytic activity of jellyfish venom is a combined effect by hemolysins, proteases, phospholipases, polypeptides and oxidizing materials. The major factors leading to hemolysis vary greatly between jellyfish species. Interestingly, functional assays of two pairs of structurally similar hemolytic proteins from *Chironex feckeri*, CfTX-1/2 and CfTX-A/B, demonstrated that CfTX-1/2 causes profound effects on the cardiovascular system of anesthetized rats, whereas CfTX-A/B elicits only minor cardiovascular effects but possesses a hemolytic activity at least 30 times greater than that of CfTX-1/2, indicating that the hemolytic proteins of jellyfish venoms have diversified structurally and functionally during evolution (*Brinkman et al., 2014*).

Our results showed that the antagonistic curves of the cations significantly right-shifted, except for $La^{3+}$, with respect to the hemolytic activity of jellyfish venom. The following four factors may also contribute to the partial inhibition of TE hemolytic

activity (except in the case of $Mn^{2+}$): (1) Direct inhibition of the non-selective cation channel. $La^{3+}$ is a well-known non-selective cation channel blocker that also functions as the best antagonist of the hemolytic activity of jellyfish venom (*Rosenthal et al., 1990*; *Bailey et al., 2005*). The potent anti-hemolytic activity of $La^{3+}$ suggests that pore formation is a major mechanism of hemolysis by jellyfish venoms (*Wang et al., 2013d*; *Ponce et al., 2016*). (2) Competition between the active cations and those essential for hemolytic activity. It was reported that the hemolytic activity of crude venom from *Carybdea alata* was dependent on the presence of divalent cations, and $Ca^{2+}$ or $Mg^{2+}$ was necessary for hemolytic activity, which might be essential for the activity of venom proteases, collagenases and $PLA_2$ (*Chung et al., 2001*; *Helmholz et al., 2007*). (3) Stabilization of the cell membrane. It has been reported that $Zn^{2+}$ and $Cu^{2+}$ influence membrane fluidity and stability (*Razin, 1972*; *Rice-Evans, 1994*) and increase resistance to hemolysis (*Lu et al., 2012*). The antioxidant effects of cations such as $Fe^{2+}$ might also hinder pore-formation via oxidation. The variability in the anti-hemolytic effects of cations suggests that the non-selective pore blocked by $La^{3+}$ contributes the most to the hemolysis of jellyfish venoms, which could be favored by other active components such as proteases, $PLA_2$ and oxidative materials.

## In vivo hemolysis and pathophysiological effect

In vivo hemolysis can occur following a jellyfish sting. Via intravenous administration of TE from jellyfish *Cyanea capillata*, we showed that in vivo hemolysis consists of the following two phases: a rapid and severe hemolysis in the first 10 min, followed by a gradual hemolysis over 3 h. Correspondingly, the indirect indexes of $K^+$ and lactic acid increased, reaching their maximum within 10 min, then recovering to levels higher than normal because of the in vivo compensation mechanism. Although the increase in hemolytic activity occurs quickly, the extent of in vivo hemolysis seemed to be much smaller than that in vitro. We have previously confirmed that the hemolytic activity of TE in diluted blood was much weaker than that in an erythrocyte suspension with the same erythrocyte ration. Both blood serum and albumin dose-dependently inhibited the hemolysis of TE (*Xiao et al., 2010*; *Wang et al., 2012*).

Despite the significant prevention from blood serum and albumin, the hemolytic activity of jellyfish is still able to damage the internal organs in direct and indirect ways. As is well known, hemolytic activity usually stems from breaks in the cell membrane and non-specific cytotoxicity, since mature mammalian erythrocytes do not have organelles. Although we do not exclude the existence of components that specifically damage important tissues and organs, the known non-specific cytotoxicity results in a basal toxicity level that can cause direct injuries to all effected organs, including the heart, liver, kidneys and lungs. The hemolytic proteins CfTX-1 and 2 (*Brinkman & Burnell, 2007*), isolated from *Chironex fleckeri* venom, possess sequence and structural similarity to the two hemolytic proteins CfTX-A and B (*Brinkman et al., 2014*; *Jouiaei et al., 2015*), also from *Chironex fleckeri* venom, which cause profound effects on the cardiovascular system but much smaller effects on erythrocytes. The relationship between these hemolytic proteins indicates the evolutional functional diversification of jellyfish hemolytic proteins

with high cardiovascular specificity. We have confirmed that the in vivo hemolysis is not strong enough to cause hypoxia in blood. The hypoxia in tissue and organs mainly stems from insufficient blood perfusion due to heart failure and vascular contraction caused by jellyfish venom.

The indirect damage of hemolysis mainly comes from released products from erythrocytes and cells. The release of large amounts of lactic acid from the erythrocytes significantly lowers the blood pH, leading to severe metabolic acidosis, which can further cause clinical manifestations such as cardiac arrhythmia, respiratory disorders and gastrointestinal symptoms. Another important factor is the elevation of blood potassium from intra-erythrocyte $K^+$ release, $H^+$–$K^+$ exchange and intracellular $K^+$ release. It has been reported that hyperkalemia is one of the main causes of cardiovascular collapse and mortality by *Chironex fleckeri* venom; this effect can be improved by zinc gluconate (*Yanagihara & Shohet, 2012*). The complete inhibition of hemolysis by the non-specific channel blocker $La^{3+}$, in addition to the partial inhibition by other cations following different mechanisms, supports the hypothesis that the lethal hemolytic mechanism is due to pore formation in cell membrane, favored over other mechanisms, and suggests an important strategy to antagonize hemolysis via pore blockage by cations, channel blockers and other antagonists.

In conclusion, we have repeated and corrected the inhibitory effects of five cations on hemolysis induced by jellyfish venom using spectrophotometric methods, and the results were further confirmed by direct erythrocyte counting under microscopy. With the exception of the transparent non-selective cation channel inhibitor $La^{3+}$, which displayed complete inhibition, the inhibitory effects of $Cu^{2+}$, $Zn^{2+}$ and $Fe^{2+}$ were right-shifted and actually weaker than those reported previously. $Mn^{2+}$ did not have any significant antagonistic effects after the correction was applied. Our results indicate that the cations, except in the case of $La^{3+}$, interfere with the absorbance of hemoglobin, which should be corrected when their inhibitory effects on the hemolytic activity of jellyfish venoms are tested. The variability in the inhibitory effects by cations supports the hypothesis that hemolysis by jellyfish venom can be attributed to the formation of non-selective cation pore complexes over other potential mechanisms, such as $PLA_2$ (*Helmholz et al., 2007*), polypeptides, proteases and oxidation. Blocking the formation of pore complexes may be a useful strategy to improve in vivo damage and mortality of jellyfish stings caused by hemolytic toxicity.

### Funding

This work was supported by the National Natural Science Foundation of China (81370833, 81470518) and the Young Scientists Fund of the National Natural Science Foundation of China (81401578). There was no additional external funding received for this study. The funders had no role in study design, data collection and analysis, decision to publish or preparation of the manuscript.

## Grant Disclosures

The following grant information was disclosed by the authors:
National Natural Science Foundation of China: 81370833, 81470518 and 81401578.

## Competing Interests

The authors declare that they have no competing interests.

## Author Contributions

- Hui Zhang conceived and designed the experiments, performed the experiments, analyzed the data, contributed reagents/materials/analysis tools, wrote the paper, prepared figures and/or tables and modified the paper.
- Qianqian Wang conceived and designed the experiments, performed the experiments, analyzed the data, contributed reagents/materials/analysis tools, reviewed drafts of the paper and modified the paper.
- Liang Xiao conceived and designed the experiments, reviewed drafts of the paper and modified the paper.
- Liming Zhang conceived and designed the experiments, reviewed drafts of the paper and modified the paper.

## Field Study Permissions

The following information was supplied relating to field study approvals (i.e., approving body and any reference numbers):

The investigation was carried out in conformity with the requirements of the Ethics Committee of the Second Military Medical University and National Institutes of Health (NIH) guide for care and use of Laboratory animals (NIH Publications No. 8023).

Jellyfish catching was permitted by the East China Sea Branch, State Oceanic Administration, People's Republic of China.

## Data Availability

The raw data has been supplied as Supplemental Dataset Files.

## Supplemental Information

Supplemental information for this article can be found online at http://dx.doi.org/10.7717/peerj.3338#supplemental-information.

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
