# Peer review of "Intervention effects of five cations and their correction on hemolytic activity of tentacle extract from the jellyfish Cyanea capillata"

_PeerJ, doi:10.7717/peerj.3338_

## Round 0.1 · original submission · Major Revisions

Your manuscript was revised by two independent reviewers that recommended it for publication. However, they have raised several comments that you have to consider/correct. I have also looked at the manuscript and, regarding, data presented in figs 4-6, I will ask you to perform VIS-spectra analysis in the presence of different ions. The spectra for Cu2+ (which is the only of the ions that was used in micromol/L) and iron (that has a mmolar concentration in blood) have to included in the revised paper. I strongly recommend you to perform at least 3 spectra for each ion. One with a concentration below de IC50, one at the IC50, and one above (ideally the highest concentration tested).

The most adequate statistical analysis to be performed is an One-way ANOVA and not t-test. Please, perform new statistical analysis.

The inhibitory effect of Cu2+, perhaps can have some implication for the treatment of potential poisoning with this cnidaria.

Reviewer 1 ·

Basic reporting

The work reported in this manuscript has been conducted to explore the effects of some cations on the hemolytic activity induced by the tentacle extract of the jellyfish Cyanaea capillata and the potential interaction between these cations and hemoglobin. It is a basic and simple research, but it is important because this information is useful for researches that use this hemolytic method in the study of cnidarian cytolysins.

Experimental design

The experimental design is appropiate.

Validity of the findings

no comment

Additional comments

In general, the manuscript is well written. However, I have some recommendations that, in my opinion, need to be considered.

Check the spaces in these lines: 37, 42, 44, 55, 62, 102, 229, 230, 234, 253, 259, 266, 273, 281, 290 and 306.

Introduction

Line 58: It is better use “cytolytic activity evaluation” instead “toxicity evaluation”, is more specific.

Line 50 – 53: the sentence is repeated.

Line 63: Revise this sentence “compared with types kinds of inhibitors”

Line 63 – 64: add reference.

Line 69 – 70: Indicate the purpose of the study, it is not necessary to describe the results in this section.

Materials and methods

Line 87: tentacle extract (TE), this abbreviation was already mentioned in the introduction section (line 70).

Line 91-92: indicate if the extract was lyophilized.

Line 94: this sections should be separated in “erythrocyte suspension” and ”hemoglobin solution”.

Results

Line 142 – 143: it is not necessary to repeat information that was described in the section of “materials and methods”.

Line 150 - 152: Add reference for this information.
Line 161: indicate statistical significance of the comparisons.

Discussion

Line 242: Revise the format of this reference (Marino et al. 2009).

Line 242 – 244: add reference of this previous study.

Line 245 – 246: add reference.

Line 285: This sentence is very important and need to be referenced.

Line 291: add reference.

Line 312 – 317: add reference.

Line 326: add reference.

References

Line 357: check the format of this reference.

There are some references that should be mentioned in the manuscript:

Helmholz et al. Comparative study on the cell toxicity and enzymatic activity of two northern scyphozoan species Cyanea capillata and Cyanea lamarckii. Toxicon.

Long and Burnett. Isolation, characterization, and comparison of hemolytic peptides in nematocyst venoms of two species of jellyfish (Chrysaora quinquecirra and Cyanea capillata). Comp. Biochem. Physiol. C.

Annotated reviews are not available for download in order to protect the identity of reviewers who chose to remain anonymous.

Reviewer 2 ·

Basic reporting

I'm not an English speaking, but in my opinion the text is clear and perfectly understandable.
I have noticed only few mistakes which I listed in the comments for authors.
In my opinion, the introduction and also the discussion are sufficiently supported by references. Of course, also other references I know could be useful to support the work, but those included are sufficient.
The figures are lear and perfecly readable. I have only one doubt which I expressed in the comments for authors.
The conclusions of the submission are coherent with results.

Experimental design

Methods are clear and well explained. Nevertheless, as I suggest in the comments for authors some explanatons would be useful.

Validity of the findings

no comment

Additional comments

This manuscript is very interesting and deals with a subject of high concern from the experimental point of view as well as for the implications in human health.
It reports interesting data about the effect of cations on hemolysis induced by Cyanea capillata extract.
I recommend to accept this manuscript subject to minor revision, as specified below.

General observations:
In materials and methods I think that Kd value is to be defined and explained (also the acronym).
In the section 2 of materials and methods the different amounts of cations are reported. I suggest the authors indicate how and because these amounts have been chosen for the experimental assessment of the inhibitory effects (on the basis of bibliographical data? On the basis of preliminary tests? Other?)
In the results section (line 158) the data about Zn2+ and Fe2+ are reported and defined as weakly suppressive. Nevertheless, by examining the graphs this seems to be doubtful. I suggest the authors explain better this statement.
The figure 6 reports the effects of cations on TE-induced hemolysis. It’s not clear because the La3+ histogram presents a value which surpasses 100%; please, explain.
Please, check all references into the text; a space is lacking before the bracket everywhere (if this is not an editorial rule).

Abstract
Line 16: I suggest to define and explain the significance of Kd

Introduction
Line 39: please, check the sentence “originating from the perioral parts”. Umbrella edge?
Line 61: jellyfish (lower-case)
Line 63: please, order the references by year of publication
Line 63: types kinds (??)
Line 71: ..... All cations, except .....

Materials and methods
Line 103: 15 mL
Line 109: hemolytic (lower case)

Discussion
Line 227: .... Mature mammalian erythrocytes .....
Line 238 and 242: the references Chung et al. 2001 and Marino et al. 2009 are prime written.
Line 253: please, add a space between “and” and “Mn2+”
Line 262: perhaps a reference is lacking. “2014” alone into brackets.
Line 270: ....... and oshem2, with respective ....
Line 309: ....mature mammalian erythrocytes ....

References
Please, check all names of organisms (genus and species( which are not italicized.

---

## Round 0.2 · Minor Revisions

Thank for sending us your review, the paper was improved considerably. However, it cannot be accepted without some additional experiments with Cu2+. In fact, since Cu change the spectrum of hemoglobin at 415 nm, the data of Cu2+ (figure 2D, in figura 3, the very low absorbance of Cu2+ can be due to the shift from 415 to 370 nm; this also applies to figure 4, 5 and 6 ) has to be confirmed at near to 370 nm. As we can see from the spectra, all concentrations of Cu2+ shifted the pear near to 420 nm to 370 nm. Consequently, authors can be either overestimating or underestimating the anti-hemolytic effect of Cu2+. In the case of iron, this applies only to the highest concentration and it will be not necessary to do additional experiments.

---

## Round 0.3 · accepted · Accept

Thank you for revising the manuscript. I strongly recommend you to include the data in the rebuttal letter (i.e., the data not included in the paper) as supplemental material.